
# Optimization shallow groundwater quality by the application of photocatalysis treatment technique in El Obour city, Egypt

El-Montser M. Seleem[1], Hossam A. El Nazer[2], Salah A.M. Zeid[1], Salman A. Salman[3], Mahmoud A. Abdel-Hafiz[1]

[1] Geology Department, Faculty of Science, Al-Azhar University, Assiut, Egypt.

[2] Chemical Sciences Department, National Research Centre, Dokki, Cairo, Egypt.

[3] Geological Sciences Department, National Research Centre, Dokki, Cairo, Egypt.

*Correspondence to*: Mahmoud A. Abdel-Hafiz (Elkarim_mahmoud@azhar.edu.eg)

**Abstract**

Collection of shallow groundwater and injecting it into the underline Miocene aquifer is a great environmental problem on the groundwater resources at El Obour city and environs. The present research work aims to investigate this water quality and validity of photocatalytic treatment of polluted water using nano-titania in presence of solar radiation. Twenty-eight representative samples were collected from various locations and their physical, chemical and microbial characteristics were determined. Bacteria analysis has been investigated for the presence of total bacterial count and indicator bacteria include total coliform, fecal coliform and fecal streptococci. The heavy metal analysis shows that more than 85% of the samples could be used for drinking in comparison with WHO specification for drinking water. The main pollutants in these samples are Cd and Pb. All the studied samples contain a viable count of heterotrophic bacteria, total coliform, fecal coliform and fecal streptococci, indicating the contamination with human and animal fecal material. The high number of indicator microorganism counts observed reflected the poor quality of water. The photocatalytic technique shows high efficiency towards the removal of more than 95% and 82% of microbial pollutants and organic residuals; respectively.

## 1 Introduction

The quality of water becomes one of the most demands in recent years due to the growing pressure of population, reclamation projects and industrial development. Therefore, the need for good water quality resources is great and comes as an essential necessity (Zeid et al., 2015; Zeid et al., 2018). The most popular water pollutants are heavy metals and bacteria; although numerous heavy metals are vital in small amounts for the normal development of the biological cycles, the majority of them become dangerous in high concentrations. In recent years, human exposure has raised dramatically as a consequence an exponential increase of their utilization in several agricultural, industrial and domestic applications (Bradl, 2002; Abdel-Hafiz, 2017).





Biological contamination of water is usually caused by the presence of living organisms, such as algae, bacteria, protozoan, or viruses. Each of these can cause distinctive problems in the water. Bacteria can be caused typhoid fever, dysentery, cholera, and gastroenteritis. Bacteria that either produce or are involved in the production of a disease called pathogenic bacteria. Pathogens that contaminate drinking water originate from the contamination of water resources, even during treatment and

within the distribution system (Gray, 2014). Therefore, their presence is determined by testing for the presence of an indicator organism, usually coliform bacteria (Adam Sigler and Bauder, 2011; Kamel, 2016; Ramadan, 2016). Microbial contamination of drinking water is a major contributor to water-borne diseases like diarrhea, nausea, gastroenteritis, typhoid, dysentery and other health-related problems (WHO, 1996; Shar et al., 2008), especially in children and persons with weak immune systems (PCRWR, 2005).

Chlorine is most widely used for drinking water treatment due to its low cost and disinfect water rapidly. However, disadvantages of water treatment with chlorine include, interact with any organic compounds in the water to produce disinfection by-products (DBPs) and to dissipate rapidly in distribution systems (WHO, 2000; Hrudey, 2009; Zhang et al., 2018). Approximately 600 DBPs are identified, by-products of this reaction include, for example some highly toxic compounds such as iodo and bromo compounds (Richardson et al., 2003; Plewa et al., 2004; Krasner et al., 2006; Deborde and Gunten,

2008), trihalomethanes (THMs) such as chloroform that is a cancer-causing agent and bromate are regulated due to their human health risk (NRC, 1977; Richardson, 2005; WHO, 2005; Murray et al., 2012). Furthermore, chlorine has limited efficacy against protozoan pathogens which an important cause of child diarrhea (Kotloff et al., 2013; Liu et al., 2016; Crider et al., 2018).

    Photocatalysis has been extensively studied and widely applied to environmental purification both in gas and liquid
phases. This could be attributed to that the photoactivated semiconductors could completely decompose (mineralize) various kinds of organic pollutants and bacteria that are refractory, toxic, and non-biodegradable, to $CO_2$ and $H_2O$ under mild conditions (room temperature and atmospheric pressure). In contrast, application of semiconductor photocatalysis by semiconductor materials has received much attention during the last three decades as a promising solution for both energy generation and environmental problems (Fuldner et al., 2010; Fuldner et al., 2011; Gagnon et al., 2016).

Titanium dioxide, as a cheap, non-toxic, and highly efficient photocatalyst, has been extensively applied for degradation of organic pollutants, for air purification, as a deodorant, for sterilization, and as air filter (Linsebigler et al., 1995). However, because of the wide band gap of titanium dioxide, only a small UV fraction of solar light (3-5%) can be utilized. Therefore, the most important and challenging issue is to develop efficient visible light sensitive photocatalysts by the modification of titanium dioxide. In the last years, anion doping of $TiO_2$ films and powders with elements like nitrogen (Gole

et al., 2004; Batzill et al., 2006).

    Since years ago, shallow groundwater table (waterlogging or saturation of soil with groundwater) raised in some places in El Obour city. It has a dangerous effect on the infrastructure in the study area and the lower groundwater aquifer quality (Zeid et al., 2015; Elnazer et al., 2017; Zeid et al., 2018). El Obour city council tried to solve this problem by setting up plenty of wells for collecting water and injecting it into the underline Miocene aquifer. These waters are expected to be





overloaded with organic, inorganic and biological pollutants which will adversely impact the underline groundwater aquifer. As a consequent of huge quantity of water loaded with fine sediments and presence of plant roots, large number of wells blocked up leading to the increase of water again (AlShahat et al., 2014; Abou Heleika and Atwia, 2015; Seleem et al., 2015; Zeid et al., 2017; Abdel-Hafiz et al., 2018). So, the scope of the present study is to find the optimal solution for the protection

of the area from waterlogging environmental impacts through, determine the water quality from the different drilled wells in El Obour city according to its content of some heavy metals and biological loads. In addition, Application of new advanced photocatalytic techniques for water treatment by using nitrogen-doped $TiO_2$ photocatalyst in the degradation and mineralizing a wide class of bacteria using the UV-visible light.

## 2 Materials and Methods

### 2.1 Sampling and preparation

Twenty-eight of water samples were collected (Fig. 1) in strong, brown glass bottles that used after sterilizing and transferred to the lab in ice box during 6 hours. The pH, TDS, and EC were determined in situ by using combined electrode (Hanna Hi93300). In the laboratory, the samples were filtered and analyzed for chemical constituents by using standard procedures (APHA 2012). Sodium ($Na^+$) and potassium ($K^+$) were determined by flame photometer (Jenway PFP7), appropriate filters

and standard curves. Total Hardness (TH) as $CaCO_3$ and chloride ($Cl^-$) were analyzed by volumetric methods. Ammonium ($NH_4^+$), sulfates ($SO_4^{2-}$) and nitrates ($NO_3^-$) were estimated by using the calorimetric technique. Calcium ($Ca^{2+}$) and magnesium ($Mg^{2+}$) were determined by using Atomic Absorption Spectrophotometer. Chromium, copper, iron, manganese and zinc were estimated by using Atomic Absorption Spectrometer (Perkin Elmer 400) at the central laboratory of the national research center (NRC), was after filtration and acidification by adding nitric acid to pH 2 (Trick et al., 2008).

The Chemical Oxygen Demand (COD) was determined by $K_2Cr_2O_7$ open reflux method (El Nazer et al., 2017). Four representative samples were selected for treatment technique.

            To determine the suitability of water for irrigation, Sodium Absorption ratio (SAR) was applied because it evaluates the sodium hazard in irrigation water (Salman and Elnazer, 2015). SAR is calculated by the following Eq. (1) (Richards, 1954) where, all values in meq/l:

$$SAR = Na^+/[(Ca^{2+} + Mg^{2+})/2]^{1/2} \tag{1}$$

The water can be classified into four categories based on SAR: excellent (SAR<10), good (10-18), doubtful (18-26) and unsuitable (>26) water for irrigation.





## 2.2 Microbial Analysis

The total bacterial count (TBC) and total count for different bacterial indicator include total coliform, fecal coliform and fecal streptococci were carried out using the most probable number (MPN) technique (APHA, 2012; Divya and Solomon, 2016; Rawway, 2016).

## 2.3 Treatment

### 2.3.1 Preparation of nitrogen doped Titania

Nitrogen-doped $TiO_2$ photocatalyst was prepared through the sol-gel hydrothermal method using titanium n-Butoxide (**TNBT**) precursor and urea. In this method, TNBT solution and urea was taken in the mole ratio 1:5. The urea solution was added drop wise to a mixture of titanium n. - butoxide and ethanol after stirring for ~ 24 hours at room temperature. It was dried at 60 ℃ and then calcined at 400 ℃ for 4 hours (Cong et al., 2007).

### 2.3.2 Photocatalytic reactions

The Photocatalytic chemical experiments were carried out by using a PHOCAT 120 W photoreactor. Aqueous dispersion of polluted water containing (1 g/l $TiO_2$) was sonicated for 5 minutes. The polluted water contacting $TiO_2$ was then irradiated using simple photoreactor using ten visible light lamps ($\lambda_{max}$ 400-700 nm) with a total power of 80 Watt. At intervals of times, 3 ml aliquots of reaction mixture were withdrawn and analyzed. When $TiO_2$ is exposed to the light, which has the photon energy equal or larger than energy gab ($E_g$) the $e^-$ in the valence band (VB) can absorb the photon and jump to the conduction band (CB), generating a positive hole in VB ($h^+_{VB}$) which are highly oxidizing. $h^+_{VB}$ can subsequently decompose organic compounds and unwanted bacteria into water and other harmless substance that disperse into the atmosphere (Gole et al., 2004; Pelaez et al., 2012).

## 3 Results and Discussion

### 3.1 Major characteristics and quality

The descriptive statistics of the studied samples are presented in Table 1. It appears from the comparison with WHO (2017) specification for drinking water that more than 90% of water samples could be used for drinking based on pH, $Mg^{2+}$ and $K^+$ values. Moreover, 70% could use for drinking based on TDS, $Cl^-$, $NH_4^+$ and $NO_3^-$; furthermore, 48% based on EC, $Ca^{2+}$, $Na^+$ and $SO_4^{2-}$ values. Ninety-three percent of water samples are very good quality and suitable for irrigation of all soils based on sodium adsorption ratio (SAR).



### 3.2 Status of some heavy metals in water

Descriptive statistical results are reported in Table 2, and compared with the world health organization (2017). The measured chromium content in water ranges from BDL to 140 µg/l. Most of the analyzed samples (89.3%) contain Cr level below maximum admissible limit for drinking water (Fig. 2a). The iron values vary from BDL up to 420 µg/l. Only two samples are out the acceptable limit (Fig. 2b). All the investigated samples have copper, manganese and zinc within the safe limits set by WHO guidelines. Thus, considered satisfactory for drinking purpose.

### 3.3 Bacteriological estimation of water

The results of total bacterial counts (TBCs) at 37 ℃ were shown in Table 3, Figs. 3 and 4, that ranged from $2.89 \times 10^4$ to $127 \times 10^4$ with average $23.5 \times 10^4$ (CFU/ml). In the present study, all of the samples were very highly contaminated with the total bacterial count in comparison with WHO (2017).

The density of total and fecal coliform is presented in Table 3 and Fig. 5. The highest values of total and fecal coliform were 2400 and 1700 (MPN/100ml), respectively; the average values recorded for total and fecal coliform was 497.4 and 358.3 (MPN/100ml), respectively. On the other hand, there are three samples are free from total and fecal coliform bacteria (<1.8 (MPN/100 ml)). This may be due to the role of soil where, bacteria are naturally filtered out by soil and rock (Ikhlil, 2009).

The fecal streptococci in water samples range from less than 1.8 to 790 MPN/100ml with an average 115.9 (MPN/100ml) (Table 3 and Fig. 5). From the bacteriological view, results showed higher counts of total bacterial counts, total coliforms, fecal coliforms and fecal streptococci, indicating the contamination of waterlogging with human and animal fecal material (Pelczar et al., 2007). The main source of fecal microorganisms in water samples are wastewater discharges (WHO, 2008). So, all water samples require effective treatment.

### 3.4 Treatment of water

Bactericidal activities of the polluted samples were evaluated and examined for total bacterial counts (TBCs) (at 37 ℃ and 22 ℃) and total coliform. In addition, two samples were examined for their content of chemical oxygen demand (COD).

As shown in Table 4, the viable bacteria were completely killed within 60 min in the N-doped TiO$_2$ suspension under visible light irradiation. As well as more than 82% of organic residuals. Doping of nitrogen into nanocrystalline TiO2 results in an extension of its light absorption into the visible region. The highly enhanced photocatalytic activity of N-TiO$_2$ may be attributed to its higher specific area, red-shifted optical absorption edge and lower optical bandgap compared to undoped-TiO2. The excellent photoactivity of N-TiO$_2$ compared with undoped one could be explained by cooperation effect between nitrogen species on increasing the photosensitivity in the visible region. Thus, N-TiO$_2$ is a cheap but a promising photocatalyst for many catalytic applications. Bandgap calculations of the doped-TiO$_2$ photocatalysts would identify the possible electron and/or energy transfer processes in this system. Previous studies suggested an electron transfer to molecular oxygen producing reactive oxygen species (Gole et al., 2004; Selvam and Swaminathan, 2012; Shi et al., 2014).



## 4 Conclusions

As a result of the present study, water is containing acceptable levels of Cr, Cu, Fe, Mn and Zn according to WHO specification for drinking water. The results of microbiological aspects indicate that all the studied samples contain a viable count of heterotrophic bacteria, total coliform, fecal coliform and fecal streptococci, reflecting the contamination with human and animal fecal material, as well as high values of COD may be due to leakage from sewage and irrigation wastewater;furthermore, industrial activities in the city. Photocatalytic treatment of polluted water using nano-titania in presence of solar radiation has shown high efficiency towards the removal of about 95% and 82% of microbial pollutants and organic residuals.

## Recommendations

The present study suggests the following recommendations be taken into account:

- Stop pumping logged water into the underline Miocene aquifer to prevent contamination of water aquifer.
- To enhance the effect of any solution, the sewage drainage and water supply networks in the city should be improved.
- Create an effective covered drainage system or drip irrigation in Orabi farms as well as green areas in El Obour city.
- Applying the photocatalytic treatment which shows the ability of photocatalytic technique towards the removal of microbial pollutants and organic residuals in safe and inexpensive ways.

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

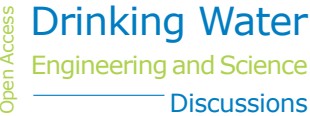

Table 1: Descriptive statistics of the physicochemical parameters, SAR and RSC of water compared with WHO (2017) specification.

| | pH | TDS mg/l | EC µS/cm | TH mg/l | $Ca^{2+}$ mg/l | $Mg^{2+}$ mg/l | $Na^+$ mg/l | $K^+$ mg/l | $SO_4^{2-}$ mg/l | $Cl^-$ mg/l | $NO_3^-$ | $NH_4^+$ | SAR |
|---|---|---|---|---|---|---|---|---|---|---|---|---|---|
| **Mean** | 7.69 | 1136.1 | 2241.6 | 314.4 | 85.3 | 33.9 | 371.8 | 5.9 | 466.5 | 273.2 | 40.0 | 0.30 | 7.88 |
| **Median** | 7.57 | 796.6 | 1536.1 | 254.1 | 69.7 | 22.5 | 247.5 | 6.0 | 223.5 | 130.6 | 16.0 | 0.16 | 6.16 |
| **SD** | 0.54 | 983.0 | 1966.2 | 222.9 | 51.9 | 32.8 | 409.4 | 3.4 | 588.6 | 379.3 | 75.5 | 0.67 | 6.79 |
| **Range** | 3.05 | 3568.6 | 7549.5 | 958.6 | 230.0 | 135 | 1260 | 12.3 | 2658 | 1428 | 346.5 | 3.65 | 23.2 |
| **Min.** | 7.18 | 318.6 | 660.9 | 85.1 | 15.0 | 5.0 | 30.0 | 0.5 | 42.0 | 31.8 | 0.0 | 0.06 | 0.74 |
| **Max.** | 8.2 | 3887.2 | 8210.4 | 1043.7 | 245.0 | 140 | 1290 | 12.9 | 2700 | 1460 | 346.5 | 3.71 | 23.9 |
| **Q1** | 7.49 | 420.6 | 818.8 | 169.5 | 50.0 | 15.0 | 73.9 | 3.1 | 83.8 | 60.0 | 3.2 | 0.15 | 2.15 |
| **Q3** | 7.77 | 1236.3 | 2350.0 | 365.3 | 100.0 | 41.3 | 453.8 | 8.7 | 723.8 | 248.3 | 28.2 | 0.21 | 12.9 |
| **WHO (2017)** | 6.5-8.5 | 1000 | 1500 | 500 | 75 | 100 | 200 | 12 | 250 | 250 | 50 | 0.2 | -- |

SD: Standard Deviation     Min. Minimum     Max. Maximum     Q1: 1st quartile     Q3: 3rd quartile



Table 2: Summary statistics of some heavy metals and compared with WHO (2017) specification.

|  | Cr μg/l | Cu μg/l | Fe μg/l | Mn μg/l | Zn μg/l |
|---|---|---|---|---|---|
| **Mean** | 23.89 | 20.71 | 73.93 | 92.86 | 78.57 |
| **Median** | 13.5 | 0.0 | 35 | 100 | 40 |
| **SD** | 31.84 | 67.87 | 98.67 | 101.57 | 196.92 |
| **Range** | 140 | 350 | 420 | 400 | 1070 |
| **Min.** | BDL | BDL | BDL | BDL | 10 |
| **Max.** | 140 | 350 | 420 | 400 | 1080 |
| **Q1** | 8 | BDL | 20 | BDL | 30 |
| **Q3** | 22.75 | BDL | 82.5 | 200 | 60 |
| **WHO (2017)** | **50** | **2000** | **300** | **400** | **3000** |

**BDL**: below detection limit



Table 3: The total bacterial count and bacterial indicators of water samples at El Obour area.

| Samples Number | Total bacterial count × 10⁴ (CFU/ml) | Bacterial indicators (MPN-index /100ml) | | |
|---|---|---|---|---|
| | at 37 ºC | TC | FC | FS |
| 1 | 4.4 | <1.8 | <1.8 | <1.8 |
| 2 | 7.3 | 1200 | 1100 | 460 |
| 3 | 3.4 | 580 | 270 | 220 |
| 4 | 4.8 | 580 | 480 | 260 |
| 5 | 69 | 1.8 | 1.8 | <1.8 |
| 6 | 47 | 260 | 250 | 3.7 |
| 7 | 11 | 270 | 250 | 3.6 |
| 8 | 50 | 320 | 310 | 61 |
| 9 | 6.0 | 310 | 260 | 1.8 |
| 10 | 32 | 380 | 310 | 1.8 |
| 11 | 7.0 | 260 | 220 | <1.8 |
| 12 | 127 | 460 | 260 | 9.1 |
| 13 | 78 | 460 | 410 | 260 |
| 14 | 11 | <1.8 | <1.8 | <1.8 |
| 15 | 11.1 | 430 | 400 | 250 |
| 16 | 4.8 | 3.7 | 3.6 | 1.8 |
| 17 | 20 | <1.8 | <1.8 | <1.8 |
| 18 | 8.7 | 940 | 840 | 790 |
| 19 | 25 | 1700 | 1200 | 250 |
| 20 | 2.98 | 270 | 220 | 120 |
| 21 | 17.8 | 460 | 260 | 250 |
| 22 | 3.1 | 2400 | 1700 | 260 |
| 23 | 9.7 | 2100 | 840 | 3.6 |
| 24 | 7.0 | 3.6 | 1.8 | 1.8 |
| 25 | 43 | 260 | 220 | 18 |
| 26 | 41 | 270 | 220 | 18 |
| 27 | 3.0 | 3.7 | 3.6 | 1.8 |
| 28 | 3.3 | 3.7 | 1.8 | <1.8 |
| Max | 127 | 2400 | 1700 | 790 |
| Min | 2.98 | <1.8 | <1.8 | <1.8 |
| Average | 23.51 | 497.4 | 358.3 | 115.9 |



Table 4: Bacterial load and COD values before and after treatment.

| | Before Treated | | | |
| :---: | :---: | :---: | :---: | :---: |
| **Sample No.** | **Total Bacterial counts $\times 10^3$** CFU/ml | | **Total coliform** MPN-index / 100ml | **COD** (mg O$_2$/l) |
| | **at 37 ºC** | **at 22 ºC** | | |
| **3** | 6.0 | 5.3 | 900 | 230.4 |
| **11** | 48 | 22 | 1500 | 960.8 |
| **19** | 12 | 8 | 150 | |
| **23** | 90 | 64 | 750 | |
| | After Treated | | | |
| **Sample No.** | **Total Bacterial counts** CFU/ml | | **Total coliform** MPN-index / 100ml | **COD** (mg O$_2$/l) |
| | **at 37 °C** | **at 22 °C** | | |
| **3** | 2 | 15 | <1.8 | 30 |
| **11** | 2 | <1.8 | <1.8 | 164 |
| **19** | 4 | <1.8 | <1.8 | |
| **23** | 7 | 2 | <1.8 | |

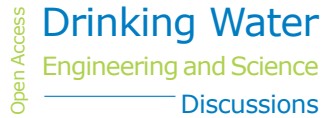

Open Access **Drinking Water**
Engineering and Science
Discussions

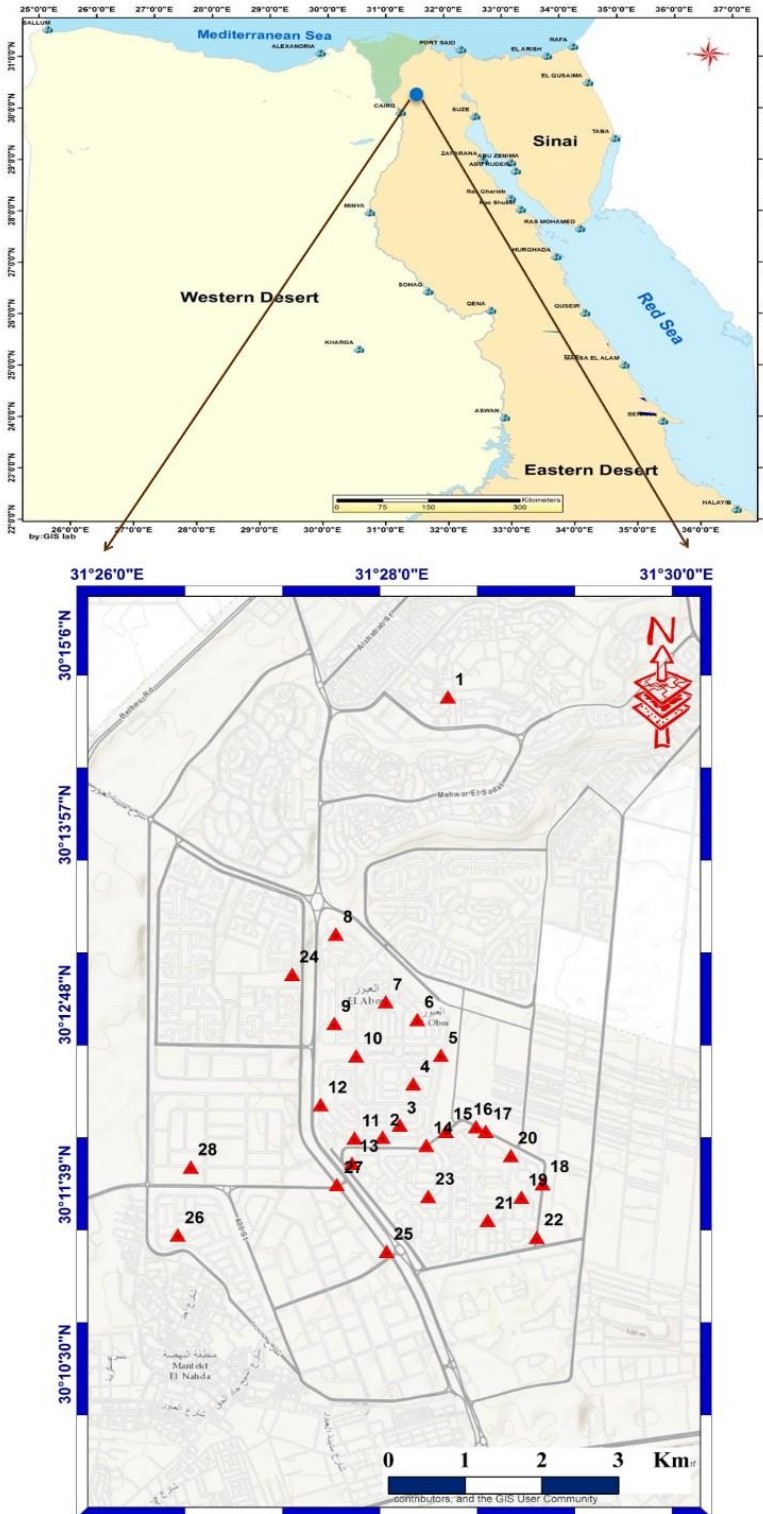

Figure 1: Location map of the collected water samples in the study area.

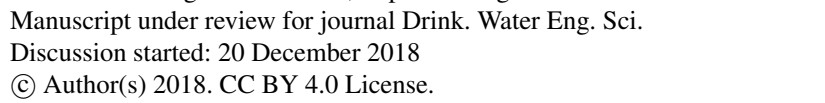

Figure 2: Comparison between WHO standard and both [(a) Cr and (b) Fe] values.

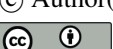

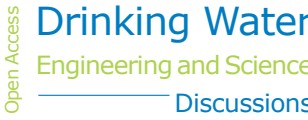

Figure 3: Total bacterial counts in water samples at the study area.

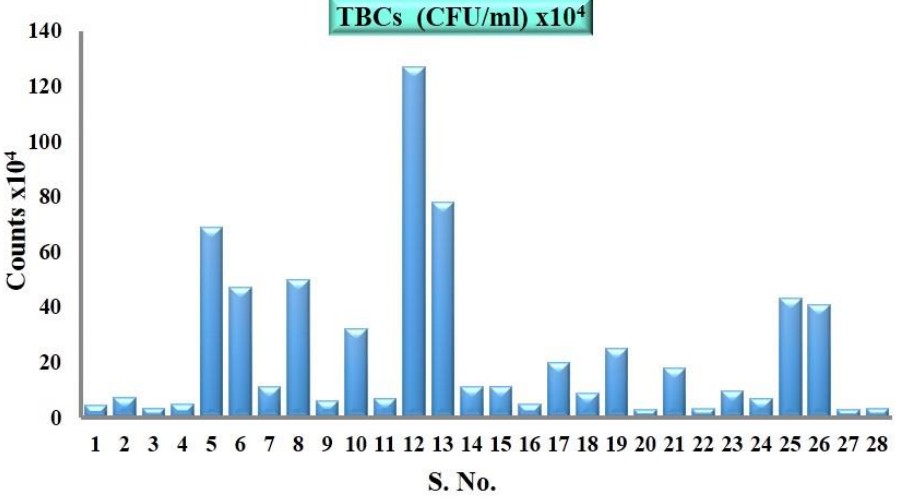

Figure 3: Total bacterial counts in water samples at the study area.

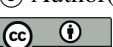



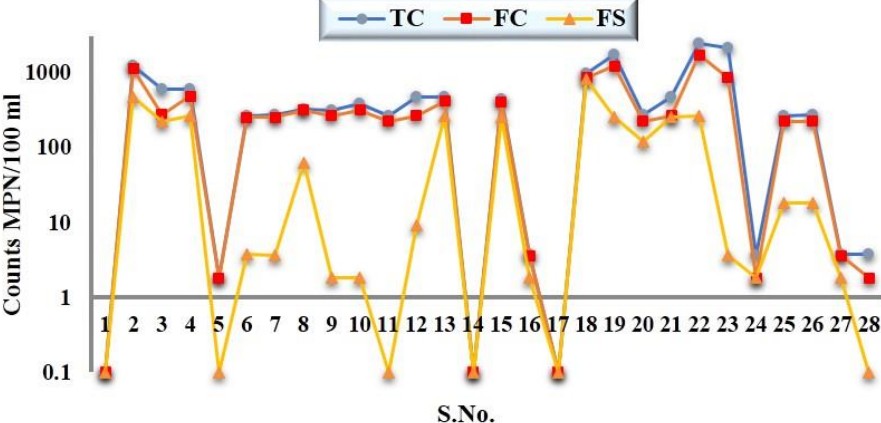

Figure 4: Bacterial indicators in water samples.