# Peer review of "Optimization shallow groundwater quality by the application of photocatalysis treatment technique in El Obour city, Egypt"

_Drinking Water Engineering and Science, 2018_

## Referee Comment (RC1) · Anonymous Referee #1 · 10 Jan 2019

This paper deals with the problem of the groundwater resources at El Obour city, Egypt due to the collection of shallow groundwater and the injection of this water into the underline Miocene aquifer. Authors found that this shallow groundwater had a viable count of heterotrophic bacteria, total coliform, fecal coliform and fecal streptococci. The aim of this research article from Introduction Section is to find the optimal solution for the protection of the area from waterlogging environmental impacts through, determine the water quality from the 5 different drilled wells. In addition, Application of advanced photocatalytic techniques for water treatment by using nitrogen-doped TiO2 photocatalyst using solar light. The paper is relevant to the scope of the journal. However, authors fail to show how to apply the photocatalytic technique for treatment of the collected

shallow groundwater. Therefore, this paper could not publish in Drink. Water Eng. Sci. journal in the current quality. This decision is according to the following comments: (1) The word "Optimization" in the title appears to be for the quality of shallow groundwater, while authors aimed to find the optimal solution for the protection of the area from this contaminated shallow groundwater (Page 3, Line 5). Thus, the title is not understandable in terms of doing the optimization for the water treatment or for the selection of the best solution. (2) In the Abstract, authors mentioned that the main pollutants in the collected 28 samples are Cd and Pb (Page 1, Line 16), while there is no information in the whole manuscript about these two heavy metals. (3) In the Introduction: # (Page 2, Line 22), why in contrast? The two sentences have the same meaning that photocatalysis is a promising solution for water treatment. # (Page 2, Line 29), the sentence for description the nitrogen doped TiO2 is not completed. # (Page 2, Line 33), The number of the wells that drilled with their dimensions can be obtained from El Obour city council and from these information the amount of water can be estimated roughly. This will give an indication for what extent of this environmental problem and help for selecting the prober water treatment technology. # The novelty of this research article is very low, especially author used known and published photocatalytic technique with nitrogen doped TiO2 in bench-scale (Reference: Cong et al., 2007). (4) Materials and Methods: # (Page 3, Line 15) The parameters TH, Cl , NH4+, SO4–, and NO3- were analyzed according to which standard method. # Standard methods is preferred to be reference for the method of determination of COD (Page 3, Line 20). # Authors referred to Cong et al., 2007, for the preparation of nitrogen doped TiO2. They used urea as a source for nitrogen with molar ratio titanium n-butoxide : urea (1:5), which is higher than was studied by Cong et al, 2007. Also, The urea is not the best source of nitrogen as found by Cong et al, 2007. In addition, no further treatment process for the obtained N-doped titania after the hydrothermal process, while authors calcined the obtained powder at 400 °C for 4 h. This will change the crystalline characteristics of the final product totally. Authors did not explain why they changed the preparation method that established by Cong et al, 2007. # In Photocatalytic Reactions Section,

holes are not the only oxidizing species in the process, what about the hydroxyl radicals and the super oxide oxygen?. (5) Results and Discussion # (Page 4, Line 23), the acceptable values of pH, Mg2+, and K+ are not sufficient to mention that more than 90% of samples could be used for drinking. This sentence must be rephrased. # (Page 5, Line 11), the Fig. number is not correct, it is 4. # (Page 4, Line 15) this paragraph did not mention the exact main source of bacterial contamination of the shallow ground water. # The visible light lamps with wavelength 400–700 nm range were used for the photocatalytic degradation process and this is not fair for showing the effect of the photocatalyst. This catalyst can be activated also in UV range from 300 nm as the solar light start from this wavelength and authors mentioned that this process can be done by solar light (see Abstract and Conclusions Sections). (6) The recommended treatment process was done in bench-scale with only 4 collected samples at fixed operation time (60 min). There is no information about the reaction kinetics. In addition, authors did not treat the problem of scaling up this heterogeneous photocatalytic process with large water volume, especially they mentioned that there are plenty of wells in the City. In addition, what about the cumbersome separation method of the nano-catalyst from the treated effluent. (7) The number of references (51) is so high for research article.

---

## Short Comment (SC1) · 11 Jan 2019

**Review report**

**For the paper entitled "Optimization shallow groundwater quality by the application of photocatalysis treatment technique in El Obour city, Egypt"**

**Manuscript Ref. dwes-2018-41**

In general, the paper is above average with important scientific merit. The paper has been written in an easily readable and understandable manner. It is well designed and structured. The experimental techniques used and the experiments are appropriate. In my opinion:

- The topic of the article is suitable for publication in **Drinking Water Engineering and Science Journal**
- The article is original with new important results.
- The article is well organized and clearly written.
- The abstract is appropriate.

For Authors

You have prepared your manuscript carefully and precisely and have put a lot of effort into output.

**I recommend accepting the manuscript for publishing after the following minor corrections:**

Page one:

Abstract, Line No(14) change The heavy metal to The heavy metals

Page four:

Line no. 13 Are $TiO_2$ or $N-TiO_2$

Page five:

Line number 24 and 27 superscript (TiO2)

Conclusion:

Change line no 6 to (Photocatalytic treatment of polluted water using N-doped nano-titania).

Recommendations:

Change the last paragraph to:

- Photocatalytic treatment shows excellent ability towards the removal of microbial pollutants as well as organic residuals in safe and inexpensive ways.

References:

Could reduce the references

Page sixteen:

Duplication of the figure caption

---

## Author Comment (AC1) · 20 Jan 2019

(1) The word "Optimization" in the title appears to be for the quality of shallow groundwater, while authors aimed to find the optimal solution for the protection of the area from this contaminated shallow groundwater (Page 3, Line 5). Thus, the title is not understandable in terms of doing the optimization for the water treatment or for the selection of the best solution. The word "Optimization" in the title is for the quality of shallow groundwater to solve the problem of water contamination and reuse of water instead of its injection into the underline groundwater Miocene aquifer, the sentence in page 3 line 5 was changed into "So, the scope of the present study is to determine

the water quality from the different drilled wells in El Obour city according to its content of some heavy metals and biological loads. Moreover, to solve the problem of water contamination and reuse of water instead of its injection into the underline groundwater Miocene aquifer through, application of advanced photocatalytic techniques for water treatment by using nitrogen-doped TiO2 photocatalyst in the degradation and mineralizing a wide class of bacteria using the UV-visible light."

(2) In the Abstract, authors mentioned that the main pollutants in the collected 28 samples are Cd and Pb (Page 1, Line 16), while there is no information in the whole manuscript about these two heavy metals. The sentence was removed from abstract

(3) In the Introduction: # (Page 2, Line 22), why in contrast? The two sentences have the same meaning that photocatalysis is a promising solution for water treatment. "In contrast, application of semiconductor photocatalysis by" Replace by "Application of semiconductor photocatalysis by"

**(Page 2, Line 29), the sentence for description the nitrogen doped TiO2 is not completed "In the last years, anion doping of TiO2 films and powders with elements like nitrogen" Replace by "In the last years, anion doping of TiO2 films and powders with elements like nitrogen has been investigated"**

**The novelty of this research article is very low, especially author used known and published photocatalytic technique with nitrogen doped TiO2 in bench-scale (Reference: Cong et al., 2007) The novelty of this research article concerns with the treatment of organics and bacteria in actual matrix in true water using photocatalytic technique in presence of TiO2 doped with nitrogen.**

**Standard methods is preferred to be reference for the method of determination of COD (Page 3, Line 20). The Chemical Oxygen Demand (COD) was determined by K2Cr2O7 open reflux method (El Nazer et al., 2017, APHA, 1998). Four representative samples were selected for treatment technique.**

\# Authors referred to Cong et al., 2007, for the preparation of nitrogen doped TiO2. They used urea as a source for nitrogen with molar ratio titanium n-butoxide: urea (1:5), which is higher than was studied by Cong et al, 2007. Also, the urea is not the best source of nitrogen as found by Cong et al, 2007. In addition, no further treatment process for the obtained N-doped titania after the hydrothermal process, while authors calcined the obtained powder at 400 ◦C for 4 h. This will change the crystalline characteristics of the final product totally. Authors did not explain why they changed the preparation method that established by Cong et al, 2007. "TNBT solution and urea was taken in the mole ratio 1:5." Replace by "TNBT solution and urea was taken in the mole ratio 5:1." (Cong et al., 2007) Replace by (Wael et al., 2015)

\# In Photocatalytic Reactions Section, C2 DWESD Interactive comment Printer-friendly version Discussion paper holes are not the only oxidizing species in the process, what about the hydroxyl radicals and the super oxide oxygen? hydroxyl radicals and the super oxide oxygen are formed in certain conditions i.e. higher pH by adding NaOH and addition of H2O2. In the present investigation NaOH or H2O2 were not added.

\# The visible light lamps with wavelength 400–700 nm range were used for the photocatalytic degradation process and this is not fair for showing the effect of the photocatalyst. This catalyst can be activated also in UV range from 300 nm as the solar light start from this wavelength and authors mentioned that this process can be done by solar light (see Abstract and Conclusions Sections). The prepared photocatalysts could absorb light from 400 to 700 nm as mentioned before in our previous article (Wael et al., 2015)

(6) The recommended treatment process was done in bench-scale with only 4 collected samples at fixed operation time (60 min). There is no information about the reaction kinetics. In addition, authors did not treat the problem of scaling up this heterogeneous photocatalytic process with large water volume, especially they mentioned that there are plenty of wells in the City. In addition, what about the cumbersome separation method of the nano-catalyst from the treated effluent. The present investigation con-

cerns with investigation of optimum conditionsfor photocatalytic treatment of collected samples in bench scale. In large scale the nano-photocatalysts would immobilized on substrates i.e Zeolites or Bentonites in fixed bed reactors since the catalysts are completely separated from the wastewater.

(7) The number of references (51) is so high for research article. Done

Please also note the supplement to this comment:
https://www.drink-water-eng-sci-discuss.net/dwes-2018-41/dwes-2018-41-AC1-supplement.pdf

---

## Author Comment (AC2) · 20 Jan 2019

Page one: Abstract, Line No(14) change The heavy metal to The heavy metals. Done

Page four: Line no. 13 Are TiO2 or N-TiO2 N-TiO2

Page five: Line number 24 and 27 superscript (TiO2) Done

Conclusion: Change line no 6 to (Photocatalytic treatment of polluted water using N-doped nano-titania). Done

Recommendations: Change the last paragraph to: • Photocatalytic treatment shows excellent ability towards the removal of microbial pollutants as well as organic residuals

in safe and inexpensive ways. Done

References: Could reduce the references Done

Page sixteen: Duplication of the figure caption The first caption was removed

Please also note the supplement to this comment:
https://www.drink-water-eng-sci-discuss.net/dwes-2018-41/dwes-2018-41-AC2-
supplement.pdf

[Figure]

**Supplement:**

**Reply to Review report**

**Dear Kamel Mahfouz**

**We are sending the deepest appreciation and respect for your kind care and attention. Great thanks for your important comments:**

Page one:

Abstract, Line No(14) change The heavy metal to The heavy metals.
Done

Page four:

Line no. 13 Are $TiO_2$ or $N\text{-}TiO_2$
$N\text{-}TiO_2$

Page five:

Line number 24 and 27 superscript (TiO2)
Done

Conclusion:

Change line no 6 to (Photocatalytic treatment of polluted water using N-doped nano-titania).
Done

Recommendations:

Change the last paragraph to:

- Photocatalytic treatment shows excellent ability towards the removal of microbial pollutants
  as well as organic residuals in safe and inexpensive ways.

Done

References:

Could reduce the references

Done

Page sixteen:

Duplication of the figure caption

The first caption was removed

---

## Referee Comment (RC2) · Anonymous Referee #2 · 21 Jan 2019

In this paper groundwater samples are taken and mainly analysed for heavy metals and pathogenic micro-organisms. Furthermore a treatment with doped TiO2 and solar light is proposed to remove the micro-organisms for drinking water purposes. The paper is scientifically of a low level, since photocatalysis is mainly interesting for the reduction of organic micro-pollutants that are not taken into account. In addition, there is no optimization performed as suggested by the title. General comments: - The objective (and knowledge gap) at the end of the introduction is not clear. Is it the purpose to use the water infiltration, for drinking or for irrigation? This determines the relevant parameters and treatment needs. - Since the rest of the paper focuses on drinking water application, the suitability for irrigation is e.g. not relevant (pg3, line 22-27) - When treatment

with photocatalysis for drinking water purposes is the main topic of the filter, heavy metal concentrations are not relevant, since they are not removed by photocatalysis (pg 5, line 1-5). - The discussion in the manuscript is poor, since the results are not related to previous work. - It is concluded that photocatalysis is a promising technology for disinfection, but it is not related to alternative treatment, e.g. UV without catalyst (which is common practice). - Recommendations section should not be part of the manuscript - Check quality of the figures, the lines between the data points have no meaning. - Check language, including tenses. - Specific comments: - Pg 1, line 11, delete "work" from abstract. - Pg1, line 19-20, how can this be concluded since "organic residuals" were not measured. - Pg 1, line 22, "became important in recent years.." - Pg 1, line 24, "popular" = "severe" - Pg 1, line 26, "consequence of an. . ." - Pg 2, line 2, delete "in the water" - Pg 2, line 2, "Pathogenic bacteria can cause. . .. . .." - Pg 2, line 3, delete sentence - Pg 2, line 7-9, avoid repetition - Pg2, line 11, "interaction" - Pg2, line 12, strange sentence, since chlorine is meant to have an residual in the network (unlike photocatalysis) - Pg 2, line 15, bromate is a consequence of disinfection with ozone and not chlorine. - Pg 2, line 21, what organic pollutants are of importance? - Pg 2, line 22-24, not relevant in this context so delete. - Pg 2, line 29, sentence should be completed - Pg 2, line 31, what is meant by "since years ago"? - Pg 2, line 31, what is the reason for raising groundwater tables? - Pag 2, line 32, delete "dangerous" - Pg 3, line 2-3, relevant in this context? - Pg 3, line 18 "estimated" = "determined" - Pg 2, line 34, "underline"? - Pg 2, line 34 "plenty of" = "many" - Pg 4, line 7, mention if the catalyst is a powder. - Pg 4, line 15-19, give "doses". - Pg 4, line 22-24, water is drinkable or not, so explain how many samples comply (and what type of treatment is needed to produce drinking water). - Pg 5, line 19, explain what needs to be removed to obtain what type of water quality. - Pg 5, line 24, which "organic residuals" are meant? - Pg 5, line 29-31, not relevant here. . . - Pg 6, line 6-8, how can this be concluded since "organic residuals" were not measured.
* * *
[Figure]

41, 2018.

---

## Short Comment (SC2) · 30 Jan 2019

Dear Reviewer Thank you very much for your efforts in reviewing this work and for your valuable comments which assist us to improve our article quality.

Replay to comments In this paper groundwater samples are taken and mainly analysed for heavy metals and pathogenic micro-organisms. Furthermore a treatment with doped TiO2 and solar light is proposed to remove the micro-organisms for drinking water purposes. The paper is scientifically of a low level, since photocatalysis is mainly interesting for the reduction of organic micro-pollutants that are not taken into account. In addition, there is no optimization performed as suggested by the title. General comments: - The objective (and knowledge gap) at the end of the introduction is not clear. The collected groundwater samples were completely analyzed (table 1&2). COD was measured in two representative samples (Table 4, page 3 line 20-21, page 5 line 22) to indicate the organic content.

Is it the purpose to use the water infiltration, for drinking or for irrigation? This determines the relevant parameters and treatment needs. - Since the rest of the paper focuses on drinking water application, the suitability for irrigation is e.g. not relevant (pg3, line 22-27) - When treatment with photocatalysis for drinking water purposes is the main topic of the filter, heavy metal concentrations are not relevant, since they are not removed by photocatalysis (pg 5, line 1-5). The study area needs drinking and irrigation water because it contains settlement, industrial and agricultural sectors. The collected water was full characterized and the concentrations of heavy metals in the studied samples nearly were the preferred limit for drinking and irrigation. So, it weren't the target of treatment (Page 5, line 2-6). Also, the studied samples are suitable for irrigation according their SAR (page 4, line 25-26). The photocatalytic treatment processes involved in degradation and mineralization of dissolved organics as well as bacteria in presence of UV-Visible irradiation.

The discussion in the manuscript is poor, since the results are not related to previous work. - It is concluded that photocatalysis is a promising technology for disinfection, but it is not related to alternative treatment, e.g. UV without catalyst (which is common practice). With respect to previous work in the study area, it is the first time to dealt with water treatment the previous studies concerned with the abstraction of water and injection of it into the underline aquifer (page 2, line 31-34). Egypt characterized by sunlight all the year so, the use of solar energy is more relevant in our study, especially we target huge volume of water. This study represents the start in this topic and more studies and construction of pilot unit is our current research topic.

- Recommendations section should not be part of the manuscript – Recommendations will be merged into Conclusion

Check quality of the figures, the lines between the data points have no meaning. – The figures will be enhanced

Check language, including tenses. Done

- Specific comments: - Pg 1, line 11, delete "work" from abstract. - Pg1, line 19-20, how can this be concluded since "organic residuals" were not measured. COD was measured in two representative samples (Table 4, page 3 line 20-21, page 5 line 22)

- Pg 1, line 22, "became important in recent years.." - Pg 1, line 24, "popular" = "severe" - Pg 1, line 26, "consequence of an: : :" - Pg 2, line 2, delete "in the water" - Pg 2, line 2, "Pathogenic bacteria can cause: : :: : ::." - Pg 2, line 3, delete sentence - Pg 2, line 7-9, avoid repetition - Pg2, line 11, "interaction" – All required corrections were done

Page 2, line 10-16, strange sentence, since chlorine is meant to have an residual in the network (unlike photocatalysis) - Pg 2, line 15, bromate is a consequence of disinfection with ozone and not chlorine. - Pg 2, line 21, what organic pollutants are of importance? - Chlorine is the applied reagent for disinfection in the Egyptian drinking water stations due to its low cost. Chlorination process can produce many harmful disinfection by-products (DBPs), as a result of interaction with organic compounds in the water (Zhang et al., 2018). Carbamate, organochlorine and organophosphorus pesticides were observed in many groundwater samples collected from the Nile aquifer in Egypt (Abdel-Shafy & Kamel, 2016). The most widespread DBPs are trihalomethanes (THMs) such as chloroform that is a carcinogenic (WHO, 2005; Murray et al., 2012). Page 2, line 22-24 was deleted

- Pg 2, line 29, sentence should be completed – The complete sentence is "In the last years, anion doping of TiO2 films and powders with elements like nitrogen has been investigated"

Pg 2, line 31, what is meant by "since years ago"? - Pg 2, line 31, what is the reason for raising groundwater tables? This problem appears recently as a result of desert

cultivation and the new settlements construction in and around the study area

- Pag 2, line 32, delete "dangerous" - Pg 3, line 2-3, relevant in this context? - Pg 3, line 18 "estimated" = "determined" - Pg 2, line 34, "underline"? - Pg 2, line 34 "plenty of" = "many" – Pg 4, line 7, mention if the catalyst is a powder. - Pg 4, line 15-19, give "doses". - Pg 4, line 22-24, water is drinkable or not, so explain how many samples comply (and what type of treatment is needed to produce drinking water). - Pg 5, line 19, explain what needs to be removed to obtain what type of water quality. – All the required corrections were done. The water can be used for drinking and irrigation after physical and chemical treatment for removing suspended, organics and bacterial content. The final catalyst is powder

Pg 5, line 24, which "organic residuals" are meant? - Pg 5, line 29-31, not relevant here: : : - Pg 6, line 6-8, how can this be concluded since "organic residuals" were not measured. Page 5, line 24, "organic residuals" replaced with "organic content" Page 5, line 29-31, were removed Page 6, line 8, "organic residuals" replaced with "organic content, respectively"

Please also note the supplement to this comment:
https://www.drink-water-eng-sci-discuss.net/dwes-2018-41/dwes-2018-41-SC2-supplement.pdf